# Inferring synaptic transmission from the stochastic dynamics of the quantal content: An analytical approach

**Zahra Vahdat[1], Oliver Gambrell[1], Jonas Fisch[2], Eckhard Friauf[2], Abhyudai Singh** [1,3,4,5]*

**1** Department of Electrical and Computer Engineering, University of Delaware, Newark, Delaware, United States of America, **2** Animal Physiology Group, Department of Biology, University of Kaiserslautern, Kaiserslautern, Germany, **3** Department of Biomedical Engineering University of Delaware, Newark, Delaware, United States of America, **4** Mathematical Sciences, University of Delaware, Newark, Delaware, United States of America, **5** Interdisciplinary Neuroscience Program, University of Delaware, Newark, Delaware, United States of America

* absingh@udel.edu

**Data availability statement:** All relevant data are within the manuscript and its Supporting information files.

## Abstract

Quantal parameters of synapses are fundamental for the temporal dynamics of neurotransmitter release, which is the basis of interneuronal communication. We formulate a general class of models that capture the stochastic dynamics of quantal content (QC), defined as the number of SV fusion events triggered by a single action potential (AP). Considering the probabilistic and time-varying nature of SV docking, undocking, and AP-triggered fusion, we derive an *exact* statistical distribution for the QC over time. Analyzing this distribution at steady-state and its associated autocorrelation function, we show that QC fluctuation statistics can be leveraged for inferring key presynaptic parameters, such as the probability of SV fusion (release probability) and SV replenishment at empty docking sites (refilling probability). Our model predictions are tested with electrophysiological data obtained from 50-Hz stimulation of auditory MNTB-LSO synapses in brainstem slices from juvenile mice. Our results show that while synaptic depression can be explained by low and constant refilling/release probabilities, this scenario is inconsistent with the statistics of the electrophysiological data, which show a low QC Fano factor and almost uncorrelated successive QCs. Our systematic analysis yields a model that couples a high release probability to a time-varying refilling probability to explain both the synaptic depression and its associated statistical fluctuations. In summary, we provide a general approach that exploits stochastic signatures in QCs to infer neurotransmission regulating processes that cannot be distinguished from simple analysis of averaged synaptic responses.

**Funding:** This work was supported by the National Institutes of Health (R01DC019268 to AS), and by BMBF (01GQ2001 to EF). The funders had no role in study design, data collection and analysis, decision to publish, or preparation of the manuscript.

**Competing interests:** The authors have declared that no competing interests exist.

## Author summary

A quantitative understanding of interneuronal communication is imperative for elucidating the information processing mechanisms in the brain. The inherent stochastic nature of neurotransmitter release at chemical synapses has been a longstanding subject of research and can be leveraged to infer quantal parameters that govern synaptic transmission in response to a train of action potentials (APs). Building on this foundation, we have developed a general stochastic model of transmitter release with time-varying parameters that define the docking of synaptic vesicles (SVs) to a finite number of docking sites in the axon terminal, and their subsequent fusion and transmitter release. The primary contribution of our study lies in providing an exact analytical derivation of the statistical distribution of the quantal content (QC), the number of SVs fusing for each AP in a train. The proposed stochastic model is employed to investigate synaptic transmission at auditory MNTB-LSO synapses in brainstem slices from juvenile mice. Our findings demonstrate that, in contrast to a simple deterministic analysis, QC fluctuations reveal a high SV release probability per AP and a rate of SV replenishment to docking sites that is high at the onset of the AP train, but decreases for subsequent APs, thereby driving short-term plasticity.

## Introduction

Action potential (AP)-triggered transmitter release is a hallmark of interneuronal communication. At a fundamental level, this communication is orchestrated via transmitter-filled synaptic vesicles (SVs) that are docked at sites in the active zone of the axon terminal, and the transmitter molecules released upon AP arrival impact the membrane potential of the postsynaptic neuron. The depletion of SVs in response to a high-frequency AP train is counteracted by their replenishment creating a dynamic equilibrium [1–3]. Recent work has unmasked diverse types of vesicle pools working sequentially or parallelly with heterogeneity among occupied docking sites [4–12], and this complexity of presynaptic processes critically shapes both the short-term and the long-term dynamics of neurotransmission in response to a train of APs [13–21].

Although several works approximate neurotransmitter release as a deterministic process [22–26], these models do not capture the variability introduced in each trial due to the inherent probabilistic nature of SV recruitment to docking sites and neurotransmitter release by AP-triggered exocytosis of SVs [27–30]. Moreover, several experimental and computational publications have argued that these stochastic effects facilitate information flow across chemical synapses [31–39], and fluctuation statistics of evoked PSCs (postsynaptic currents) provides robust estimates of presynaptic model parameters [40,41].

There is a rich tradition of using fluctuation statistics arising from the inherent stochastic nature of transmitter releases to infer synaptic parameters [42]. The classical approach for doing this is the variance-mean analysis, where the parabolic relationship between the variance and mean amplitudes of postsynaptic responses is fitted to corresponding statistics obtained from data by varying the release probability [43–45]. The variance-mean analysis has been applied to infer quantal parameters at diverse central synapses [45–47] and the neuromuscular junction [48,49]. A key assumption of this approach is that synaptic responses remain temporally stable for sufficiently long durations to give reliable estimates of the mean and variance. Building on this tradition of leveraging fluctuations, several recent methods

generalize the approach to consider short-term plasticity in synaptic responses by finding parameters that maximize the likelihood of observing data in response to a train of APs given an underlying stochastic model of transmitter release [50].

In prior work, we have used the formalism of Stochastic Hybrid Systems to develop mechanistic models of neurotransmission investigating how diverse noise mechanisms shape the statistics of SV counts [51,52] and their corresponding impact on postsynaptic AP firing times [53]. Here we generalize these models to consider probabilistic docking and undocking of SVs at a fixed number of docking sites. Docked and primed SVs represent the readily releasable pool (RRP) of SVs, and each AP triggers probabilistic SV fusion and neurotransmitter release. A key feature of the model is that all these probabilities can *vary arbitrarily over time*, thereby capturing diverse response dynamics, including synaptic facilitation and depression. These transient parameters reflect a variety of physiological processes during high-frequency stimulation, such as buildup in calcium concentrations in the axon terminals or depletion of upstream SV recycling pools that lead to reduced recruitment of SVs to docking sites.

The key analytical contribution of this paper is the *exact* analytical solution for the transient statistical distribution of the quantal content (QC), defined as the number of SV fusion events per AP, for such a general stochastic model of synaptic transmission with time-varying parameters. When APs arrive deterministically (i.e., at fixed time points) in the axon terminals, the transient QC distribution follows a binomial distribution. In contrast, deviations from the canonical binomial behavior occur when APs arrive stochastically. The statistical dispersion in QC (as quantified by the QC Fano factor, i.e., the variance divided by the mean) is shown to be a monotonically decreasing function of the mean QC, implying higher statistical fluctuations for stronger synaptic depression. Intriguingly, our analysis shows that *different parameter regimes, that are otherwise indistinguishable from their average QC dynamics, yield contrasting QC Fano factor profiles*. Thus, a systematic statistical study of transient QC fluctuation from electrophysiologically obtained data can be a valuable tool to infer processes regulating neurotransmission. Finally, we investigate the extent of steady-state QC fluctuations as a function of model parameters and also derive an exact analytical expression for the QC auto-correlation function (i.e., the Pearson correlation coefficient between two QCs separated by a given number of stimuli).

The applicability of our modeling results is illustrated by single-cell recordings of postsynaptic responses for 3,000 stimuli (50-Hz stimulation for 1 min) at inhibitory glycinergic MNTB-LSO synapses in the auditory brainstem. Classical parameter estimation approaches, such as the method of Elmqvist and Quastel (EQ) [54], or simply fitting the synaptic depression and steady-state performance, predict low constant values for refilling probability and release probability. However, these parameters significantly overestimate the magnitude of statistical fluctuations in QC and anticorrelations between successive QCs as seen in the electrophysiological data. The combination of our analytical results with the experimentally observed fluctuation statistics reveals a dramatically different picture of high release probability and high refilling probability at these robust auditory synapses involved in sound localization. Both are critical for sustained neurotransmission at high frequency and fidelity. We begin with a detailed description of the stochastic model and highlight key underlying assumptions.

## Methods

### Stochastic formulation of neurotransmission

We consider that APs arrive in the axon terminal with a given frequency $f$ at deterministic times $\{0, 1/f, 2/f, ...\}$. The stochastic dynamics of QC occur as per the following rules, with model parameters summarized in Table 1:

- There are $M \in \{1, 2, ...\}$ docking sites in the active zone, and each site can be either empty or occupied by an SV.
- We assume that at the start of stimulation, when the first AP arrives, each docking site is occupied with probability $p_1$.
- Upon arrival of the $i^{th}$ AP, where $i \in \{1, 2, ...\}$, each docked SV has a probability $p_{r,i}$ of fusing and releasing the neurotransmitter by fast exocytosis. Upon SV fusion, the corresponding docking site is assumed to instantaneously transit to an empty state. We refer to $p_{r,i}$ as the *release probability* and this is assumed to be an arbitrary function of $i$, reflecting transient changes in its value in response to calcium buildup in the axon terminal.
- Between successive APs $i$ and $i + 1$, each empty site can be reoccupied with probability $p_{d,i}$. We refer to $p_{d,i}$ as the *refilling probability* and it is also an arbitrary function of $i$.
- Motivated by recent observations of "transient docking" [55,56], we also consider the scenario where each occupied site can become empty due to SV undocking with probability $p_{u,i}$ between APs $i$ and $i + 1$.
- Sites are assumed to be *identical* in terms of their refilling/undocking/release probabilities and operate *independently* of each other.

**Table 1. Parameters used in the stochastic model for synaptic transmission.**

| Parameter | Description |
|---|---|
| $M$ | Number of docking sites for synaptic vesicles (SVs) in the axon terminal. |
| $f$ | Frequency of action potential (AP) arrival at the axon terminal. |
| $n_i$ | Random variable representing the number of docked readily releasable SVs just before the arrival of the $i^{th}$ AP for $i \in \{1, 2, ...\}$. |
| $p_i$ | The probability of a docking site being occupied by an SV just before the arrival of the $i^{th}$ AP. |
| $\bar{p}$ | The steady-state value of $p_i$ also referred to as the normalized synaptic depression. |
| $p_{r,i}$ | The probability per docked SV to fuse and release neurotransmitter upon arrival of the $i^{th}$ AP. Represented as $p_r$ when $p_{r,i}$ is constant across APs. |
| $p_{u,i}$ | The probability per site of a docked SV undocking between the $i^{th}$ and $i + 1^{th}$ AP. Represented as $p_u$ when $p_{u,i}$ is constant across APs. |
| $p_{d,i}$ | The probability of an empty site getting reoccupied (or refilled) by an SV between the $i^{th}$ and $i + 1^{th}$ AP. Represented as $p_d$ when $p_{d,i}$ is constant across APs. |
| $b_i$ | Random variable representing the quantal content (QC) - the number of SVs fusing upon arrival of the $i^{th}$ AP. |
| $\langle b_i \rangle$ | Average QC for the $i^{th}$ AP. |
| $FF_i$ | The QC Fano factor (variance divided by the mean) quantifying the statistical dispersion in QC $b_i$. |
| $FF$ | The steady-state value of $FF_i := \lim_{i \to \infty} FF_i$. |
| $\rho$ | The steady-state Pearson correlation coefficient between successive QCs. |
| $FF_i^e$ | The Fano factor of evoked PSC (postsynaptic current) peak amplitude upon arrival of the $i^{th}$ AP. |
| $FF^e$ | The steady-state value of $FF^e := \lim_{i \to \infty} FF_i^e$. |
| $\rho^e$ | The steady-state Pearson correlation coefficient between successive evoked PSC amplitudes. |
| $\langle c \rangle$ | The average quantal size. |
| $CV_q$ | The coefficient of variation in quantal size. |

We refer the reader to Appendix A in S1 File, where probabilities $p_{d,i}$ and $p_{u,i}$ are directly linked to the kinetic rates of SV docking and undocking at individual sites. These probabilities are also linked to AP timing and thus change with stimulation frequency. A schematic of the model is shown in Fig 1A together with a sample realization in Fig 1B. It is important to point out that the proposed model with time-varying probabilities is a generalization of the stochastic Tsodyks-Markram model classically used to capture short-term synaptic plasticity [50,57–60]. While several of these models do not consider SV undocking, this assumption has been relaxed in recent work [61].

## Results

### Transient distribution of quantal content

Having defined the stochastic model in the previous section, we first present our main theoretical result quantifying the *exact transient* statistical distribution of QC.

Given an initial probability $p_1$ of a docking site being occupied by an SV and a sequence of time-varying probabilities $p_{r,i}, p_{u,i}, p_{d,i}$ for $i \in \{1, 2, ...\}$, then the number of docked SVs $\boldsymbol{n}_i$ just

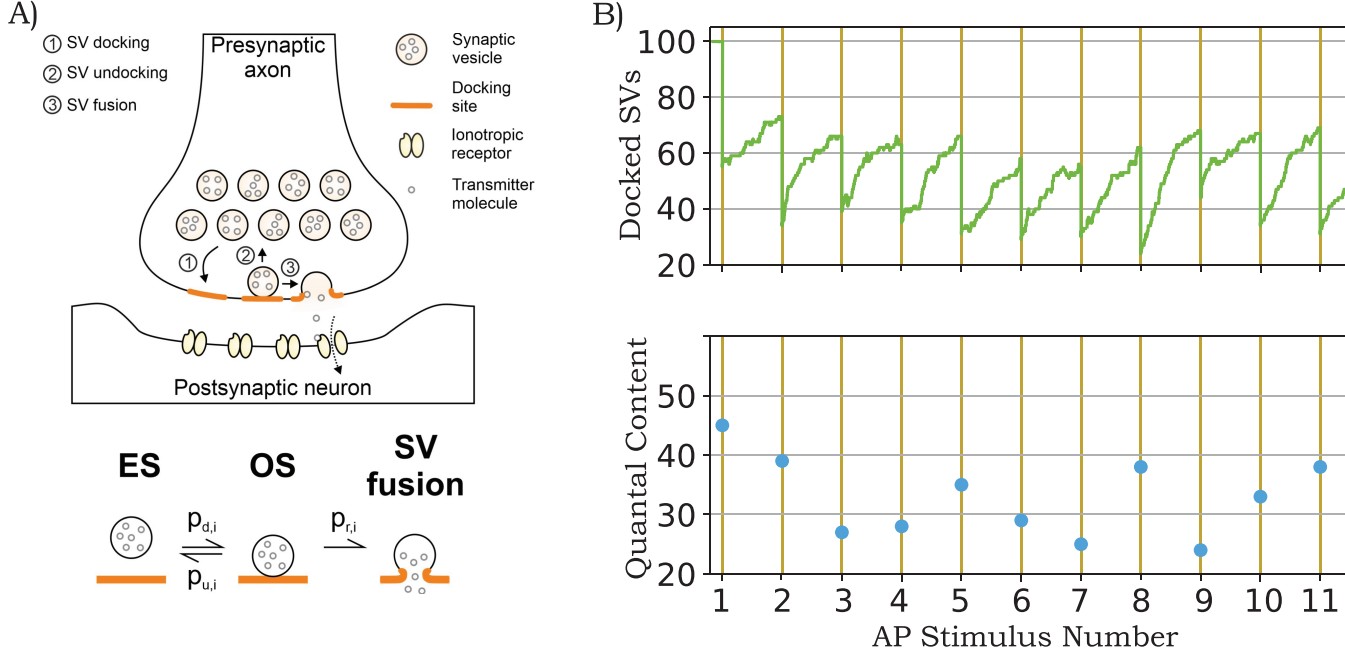

**Fig 1. Schematic of a chemical synapse and sample realizations of its corresponding stochastic model**. A) The process of SV docking and undocking in the active zone of the axon terminal, and the evoked release of neurotransmitter molecules. The lower panel shows the different time-varying probabilities $p_{d,i}$, $p_{u,i}$, $p_{r,i}$ related to the $i^{th}$ AP, where $i \in \{1, 2, ...\}$, that govern the reversible transitions between an empty site (ES) and an occupied site (OS) upon SV docking/undocking, and SV fusion (see text for details). B) A sample stochastic realization of the model showing a buildup in the number of docked SVs between successive APs, and a reduction in docked SV numbers from fusion and neurotransmitter release in response to APs (top). The corresponding quantal content (QC) – the number of SV fusion events per AP – is shown in the bottom plot. The number of docking sites is assumed to be $M = 100$, and all sites are occupied with SVs at the start of the AP train. Each docked SV has a constant release probability $p_r = 0.5$, and drawing from a binomial distribution results in the first QC to be 45. This drops the number of docked SVs to 55 after the first AP, and docked SVs replenish till the arrival of the second AP. The docked SV dynamics in the inter-stimulus interval is governed by kinetic rates that are chosen so as to result in refilling and undocking probabilities $p_d = 0.4$ and $p_u = 0.1$, respectively. (Appendix A in S1 File).

before the arrival of the $i^{th}$ AP follows the binomial distribution

$$\text{Probability}\{\boldsymbol{n}_i = j\} = \binom{M}{j} p_i^j (1 - p_i)^{M-j}, \quad j = \{0, \ldots, M\} \tag{1}$$

corresponding to each of the $M$ sites with probability $p_i$. This probability $p_i$ is the solution to the recursive equation

$$p_{i+1} = p_i(1 - p_{r,i})(1 - p_{u,i}) + (1 - p_i(1 - p_{r,i}))p_{d,i}. \tag{2}$$

The number of SVs fusing to release neurotransmitter $\boldsymbol{b}_i$ (i.e. the QC) in response to the $i^{th}$ AP follows the binomial distribution

$$\text{Probability}\{\boldsymbol{b}_i = j\} = \binom{M}{j} (p_i p_{r,i})^j (1 - p_i p_{r,i})^{M-j}, \quad j = \{0, \ldots, M\}, \tag{3}$$

where the binomial coefficient $\binom{M}{j}$ is defined as

$$\binom{M}{j} := \frac{M!}{j!(M-j)!}. \tag{4}$$

The detailed proof can be found in Appendix B in S1 File. While this result is for deterministic arrivals of APs, it can be easily generalized to consider the time between APs following an independent and identically distributed (i.i.d.) random variable, in which case $\boldsymbol{b}_i$ no longer follows a binomial distribution (Appendix C in S1 File). From this theorem, the average QC is given by

$$\langle \boldsymbol{b}_i \rangle = M p_i p_{r,i} \tag{5}$$

where angular brackets $\langle \ \rangle$ denote the expected value of random variables and random processes.

The statistical dispersion in QC at the $i^{th}$ stimulus is quantified using the Fano factor

$$FF_i := \frac{\langle \boldsymbol{b}_i^2 \rangle - \langle \boldsymbol{b}_i \rangle^2}{\langle \boldsymbol{b}_i \rangle} \tag{6}$$

defined as the variance divided by the mean. For a binomial random variable $\boldsymbol{b}_i$

$$FF_i = 1 - p_i p_{r,i} \leq 1, \tag{7}$$

and is always upper-bounded by one. *An important consequence of this result is that one can directly connect the mean QC for the $i^{th}$ AP to the corresponding statistical fluctuations in QC for any arbitrary time-varying probabilities $p_{r,i}, p_{u,i}, p_{d,i}$.* From (5) and (7), one can rewrite the QC Fano factor as

$$FF_i = 1 - \frac{\langle \boldsymbol{b}_i \rangle}{M}. \tag{8}$$

$FF_i$ increases with decreasing $\langle \boldsymbol{b}_i \rangle$, *implying that a stronger reduction in QC is associated with increased randomness in evoked release* (Fig 2). As $\langle \boldsymbol{b}_i \rangle \to 0$, $FF_i \to 1$, where a Fano factor of one corresponds to Poisson-distributed $\boldsymbol{b}_i$.

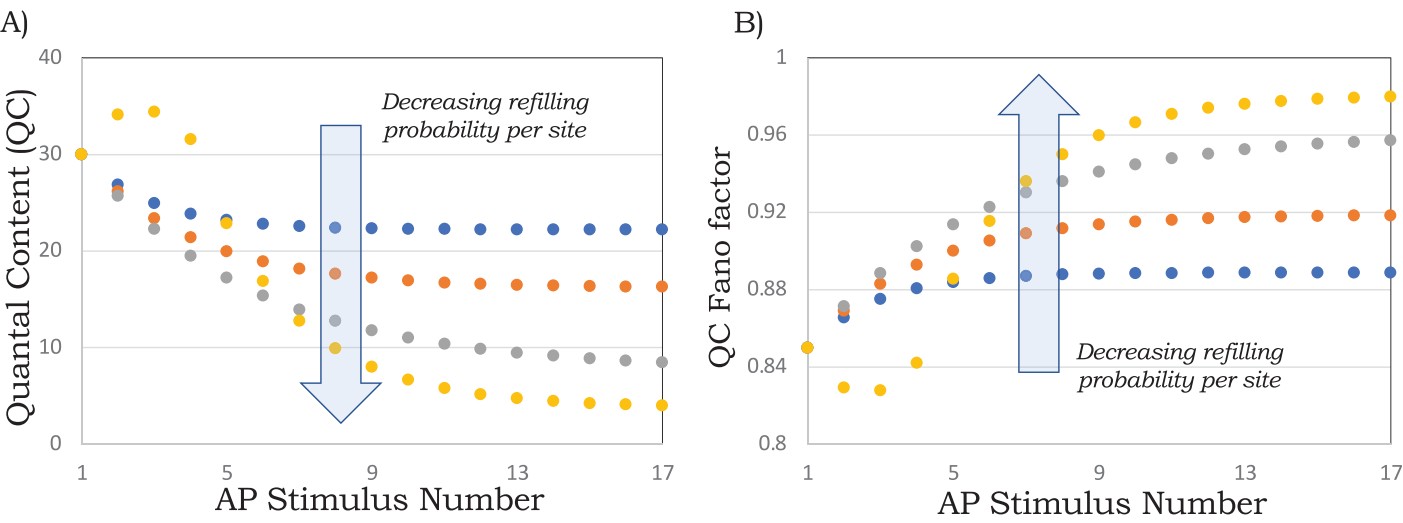

**Fig 2. Transient reduction in quantal content (QC) is associated with an increased QC Fano factor.** A) The average QC, i.e., the average number of synaptic SVs fusing per AP as predicted by Eqs (5) and (10) for a constant release probability $p_r = 0.15$ (blue, orange, and gray dots), and undocking probability $p_u = 0$. To capture synaptic facilitation (yellow dots) we also consider a time-varying release probability: $p_{r,1} = 0.15$ (first stimulus), $p_{r,2} = 0.2$ (second stimulus), $p_{r,3} = 0.25$ (third stimulus) and $p_r = 0.3$ for the fourth and all remaining stimuli. The number of docking sites $M = 200$ are all assumed to be filled upon arrival of the 1st AP. The per-site SV refilling probability is $p_d = 0.3$ (blue dots), 0.15 (orange dots), 0.05 (gray dots), or 0.02 (yellow dots). B) The corresponding QC Fano factor $FF_i$ over time as predicted by (7).

An important special case is when refilling/undocking/release probabilities take constant values independent of the AP number

$$p_d := p_{d,i}, \quad p_u := p_{u,i}, \quad p_r := p_{r,i}, \quad \forall i \in \{1, 2, \ldots\}. \tag{9}$$

Then, solving the recurrence Eq (2) yields

$$p_i = \frac{\left(p_d + (1 - p_r)^{i-1}(1 - p_d - p_u)^{i-1}\left(-p_d + p_1\left(p_d + p_u + p_r(1 - p_d - p_u)\right)\right)\right)}{p_d + p_u + p_r(1 - p_d - p_u)}, \tag{10}$$

and the corresponding statistical fluctuations in QC are binomially distributed with mean $M p_i p_r$ and Fano factor $1 - p_i p_r$. When $p_u = 0$ and $p_1 = 1$, (10) reduces to

$$p_i = \frac{\left(p_d + p_r(1 - p_r)^{i-1}(1 - p_d)^i\right)}{p_d + p_r(1 - p_d)}, \quad \forall i \in \{1, 2, \ldots\}. \tag{11}$$

Taking the limit $i \to \infty$ in (10) we obtain at steady-state

$$\bar{p} := \lim_{i \to \infty} p_i = \frac{p_d}{p_d + p_u + p_r(1 - p_d - p_u)}, \tag{12}$$

$$= \frac{p_d}{p_d + p_r - p_r p_d} \quad \text{when} \quad p_u = 0. \tag{13}$$

Fig 2A shows the mean QC $M p_i p_r$ as given by (10), and the depression in the synaptic response is exacerbated with decreasing per site refilling probability $p_d$. Furthermore, as predicted by (8), the stochasticity in QC increases over time (Fig 2B).

It is interesting to point out that the dynamics of the mean QC can be explained by alternative parametric sets that have dramatically different predictions on the QC Fano factor (Fig 3). To see this, consider the gray trace in Fig 2A that shows the transient reduction in the average QC corresponding to constant and low probabilities $p_r = 0.15$ and $p_d = 0.05$. The corresponding Fano factor over time is shown in Fig 2B and repeated in Fig 3 (top-most curve) for contrasting purposes. Now consider an alternative scenario with a high probability of release. Considering the extreme case of $p_r = 1$, the average QC is given by

$$\langle \boldsymbol{b}_i \rangle = M p_{d,i}, \tag{14}$$

implying that the same decrease in $\langle \boldsymbol{b}_i \rangle$ can be a result of a decreasing refilling probability $p_{d,i}$ due to depletion of SV pools upstream of the RRP. Note that in this case of $p_r = 1$, the number of docking sites will have to be much lower than in the former case of $p_r = 0.15$ and $p_d = 0.05$ to have the same QC for the first stimulus. However, this scenario predicts a Fano factor profile that starts very low and sharply increases over time (middle curve in Fig 3). Finally, consider a third scenario where $p_r = p_d = 1$ (i.e., all empty sites get occupied in the inter-stimulus interval and each vesicle is released with probability one upon AP arrival). In this case, the decrease in QC can be potentially explained by a reduction in the number of docking sites $M$ due to impaired access to sites. Note that the predicted noise (7) is independent of $M$, and in

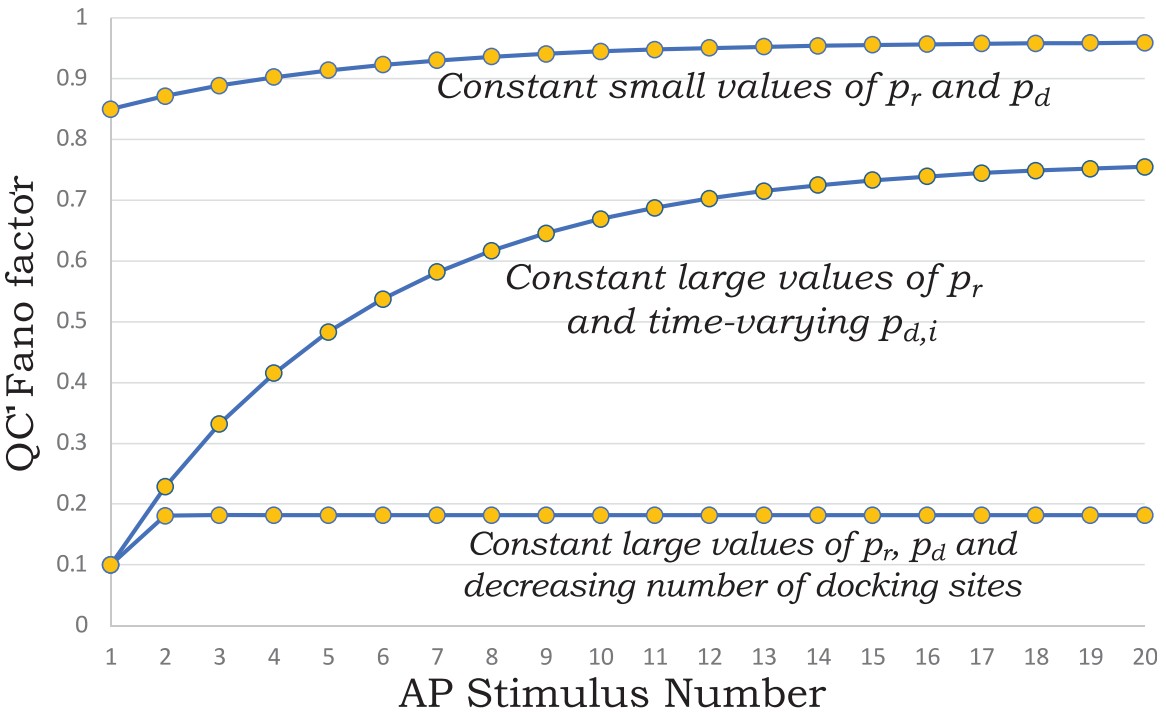

**Fig 3. Alternative parameter regimes with identical mean transient QC yield contrasting QC fluctuation statistics.** Different predictions for the QC Fano factor, each resulting in the same mean QC corresponding to $p_r = 0.15$ and $p_d = 0.05$ in Fig 2 (the bottom-most, yellow line). The top line corresponds to the Fano factor $FF_i$ predicted for $p_r = 0.15$ and $p_d = 0.05$ from (7). The middle curve is obtained from (7) with a high release probability ($p_r = 0.9$) and a corresponding time-varying refilling probability $p_{d,i}$ to get the same mean synaptic depression. The bottom curve corresponds to (7) with parameters $p_r = p_d = 0.9$, and here the same mean synaptic depression occurs due to a reduction in the number of docking sites $M$. In all cases, the undocking probability is assumed to be zero ($p_u = 0$) and each docking site is occupied at the beginning of the AP train ($p_1 = 1$).

this case, $FF_i$ is predicted to be low throughout (bottom-most curve in Fig 3). Consistent with previous work [43,50], these hypothetical examples emphasize that QC noise statistics contain useful signatures providing additional insights into the mechanisms underlying synaptic depression.

## Steady-state QC fluctuation statistics

Assuming that the probabilities $p_{r,i}, p_{u,i}, p_{d,i}$ reach their respective constant values $p_r, p_u, p_d$, we investigate the steady-state QC statistics. Our results from the previous section imply that the steady-state QC distribution is binomial with mean $M\bar{p}p_r$ and Fano factor $1 - p_r\bar{p}$ [62], where using (12) we obtain

$$FF := \lim_{i \to \infty} FF_i = \frac{p_d + p_r + p_u(1 - p_r) - 2p_r p_d}{p_d + p_r + p_u(1 - p_r) - p_r p_d} \tag{15}$$

that reduces to

$$FF = \frac{p_d + p_r - 2p_r p_d}{p_d + p_r - p_r p_d} \tag{16}$$

when $p_u = 0$. If either of the two probabilities is low (i.e., $p_d \ll 1$ or $p_r \ll 1$) then $FF \approx 1$ as illustrated in Fig 4A. However high values for both these probabilities result in a low $FF$, with

$$FF \approx 2 - p_d - p_r \quad \text{if } p_r \approx 1 \text{ and } p_d \approx 1. \tag{17}$$

As $FF$ in (16) is a function of both probabilities, it by itself cannot be used to infer $p_r, p_d$ independently. However, note from (16) that

$$p_r \geq 1 - FF \text{ and } p_d \geq 1 - FF, \tag{18}$$

and thus $FF$ provides a useful lower bound of both these parameters.

It is well known that using correlations between successive quantal contents can significantly improve parameter estimates [50,58,63]. Motivated by that, we report the steady-state Pearson correlation coefficient between successive QCs for our stochastic model as

$$\rho := \lim_{i \to \infty} \frac{\langle \boldsymbol{b}_{i+1}\boldsymbol{b}_i \rangle - \langle \boldsymbol{b}_i \rangle^2}{\langle \boldsymbol{b}_i^2 \rangle - \langle \boldsymbol{b}_i \rangle^2} = -\frac{p_d(1 - p_r)p_r(1 - p_d - p_u)}{p_r + p_d + p_u(1 - p_r) - 2p_d p_r} \leq 0, \tag{19}$$

where for $p_u = 0$

$$\rho = \frac{p_d p_r (1 - p_r)(1 - p_d)}{2p_d p_r - p_r - p_d} \leq 0. \tag{20}$$

Note that this correlation is predicted to be always non-positive, i.e., a higher-than-average QC would result in the next QC being lower-than-average due to SV depletion. We refer to Appendix D for proofs and generalization of these results to correlations between $\langle \boldsymbol{b}_i \rangle$ and $\langle \boldsymbol{b}_{i+\ell} \rangle$, where $\ell \in \{1, 2, \ldots\}$. Moreover, Appendix E extends these results to account for quantal size fluctuations.

As illustrated in Fig 4B, $\rho$ is low if either of the probabilities $p_r$ or $p_d$ takes values close to zero or one, and stronger anticorrelation is seen at intermediate values of both probabilities. The minimum value of $\rho = -0.125$ is attained at $p_r = p_d = 0.5$. For a given value of $p_d$, $\rho$

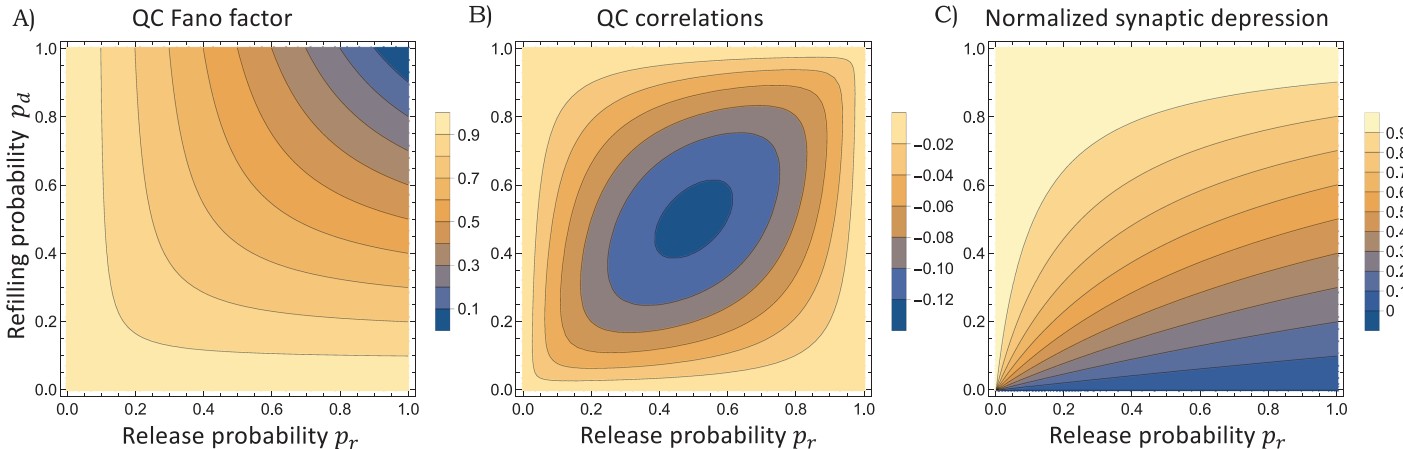

**Fig 4. Normalized synaptic depression and QC fluctuation statistics as a function of release and refilling probabilities**. A) Plots of the steady-state QC Fano factor as given by Eq (16). B) Steady-state correlation $\rho$ between successive QCs as given by Eq (20). C) Normalized synaptic depression assumed to be equal to $\bar{p}$ in Eq (13) as a function of the release and refilling probabilities. From panel B, one can see that if both probabilities $p_r$ and $p_d$ are simultaneously high or simultaneously low, this leads to uncorrelated QCs. However, the two scenarios make contrasting predictions on the Fano factor in panel A. In particular, high probability values lead to a Fano factor close to zero, whereas low probability values result in a Fano factor close to one.

varies non-monotonically with respect to the release probability (Fig A in S1 File) attaining a minimum value when

$$p_r^* = \frac{p_d^2}{(1 - p_d)^2 + p_d^2}. \tag{21}$$

Finally, Fig 4C plots the normalized synaptic depression defined as the steady-state average QC normalized by its corresponding value in response to the first stimulus. This is defined by the ratio

$$\frac{\bar{p}p_r}{p_1 p_{r,1}} \geq \bar{p}. \tag{22}$$

Recall that $p_1$ is the probability of a docking site being occupied at the first AP, and $p_{r,1}$ is the corresponding release probability. In many cases $p_{r,1}$ is much lower than its steady-state value $p_r$ due to $Ca^{2+}$ buildup in the presynaptic axon terminal, and the ratio (22) is lower bounded by $\bar{p}$ as given by (13). Fig 4C plots the normalized synaptic depression $\bar{p}$ as given by (13) assuming $p_1 = 1$ and $p_{r,1} = p_r$, and is sensitive to the refilling probability with

$$\lim_{p_d \to 0} \bar{p} = 0 \quad \text{and} \quad \lim_{p_d \to 1} \bar{p} = 1. \tag{23}$$

As illustrated next, combining knowledge of normalized synaptic depression and QC fluctuation statistics from electrophysiological data with formulas presented here provides an effective tool to infer model parameters.

## MNTB-LSO synapses: An experimental case study

We applied the mathematical results developed here to the study of neurotransmission in the auditory system. Auditory neurons can fire APs at high rates and are able to do so continuously to enable sound localization as well as object and speech recognition in noisy environments [64–70]. Specifically, we used published data from electrophysiological recordings

in juvenile mouse brain slices of the inhibitory glycinergic connection between the medial nucleus of the trapezoid body (MNTB) and the lateral superior olive (LSO) in the medullary brainstem (hereafter referred to as MNTB-LSO synapses). This connection plays a role in sound localization by analyzing interaural intensity differences [71–74].

Fig 5A shows the QC estimation during a whole-cell patch-clamp recording of a single LSO neuron when MNTB axons were stimulated at 50 Hz for 1 min (AP train with 3000 stimuli) as taken from [69] and QC data provided in S2 File. As shown in Fig 5B (a close-up of Fig 5A for AP numbers 1-20), the normalized QC reaches a steady state after an initial decrease. The reader is referred to [69] for experimental details; the QC is obtained by dividing the peak amplitude of evoked PSCs (postsynaptic current) by the average spontaneous PSC peak amplitude in the same neuron. The spontaneous PSCs followed a Gaussian distribution with a mean of 22 pA and were found to be the same before stimulation and at the end of the 50 Hz AP train for 1 min [69]. Steady-state statistics are quantified using QCs from stimulus number 10 to 3000. We focus on two steady-state metrics: the Fano factor (*FF*; variance divided by mean) and the Pearson correlation coefficient $\rho$ between consecutive QCs.

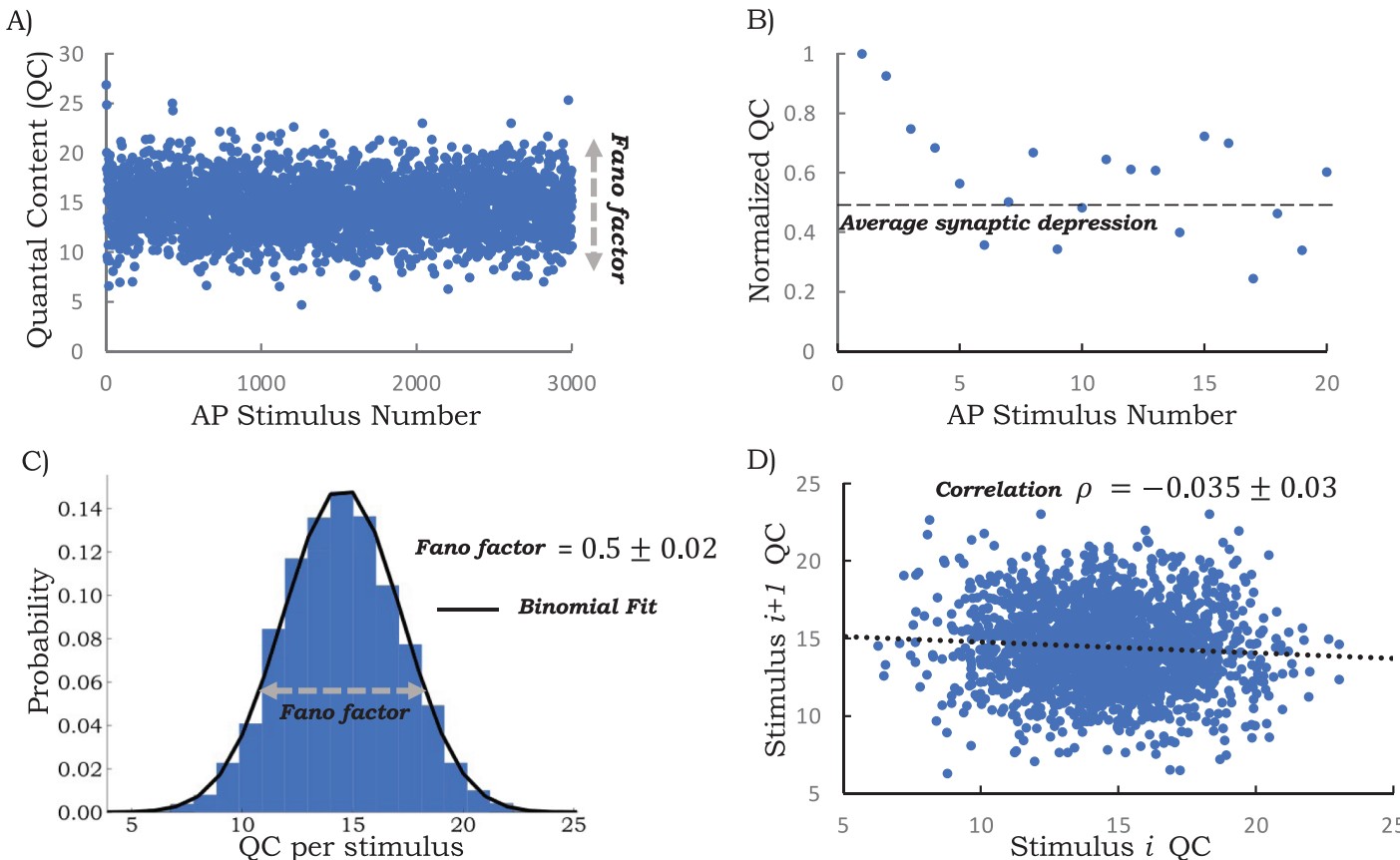

**Fig 5. Fluctuation statistics of the quantal content (QC) for the auditory MNTB-LSO synapses**. A) Results of a whole-cell patch-clamp recording from a single LSO neuron at a 50-Hz challenge for 1 min (3000 stimuli) as obtained from [69]. Each point represents the QC after a single stimulus pulse. B) Normalized QC values to the first 20 stimuli show the initial depression behavior and the subsequent average synaptic depression level. Values are normalized to the first stimulus QC. C) Steady-state QC distribution as obtained using QCs from stimulus numbers 10 to 3000 together with a fit to a binomial distribution. The steady-state QC Fano factor is obtained as *FF* = 0.5 ± .02, where ± denotes the 95% confidence intervals as obtained by bootstrapping. D) The scatter plot between successive QCs from AP number 10 to 3000 shows a weak negative correlation with a Pearson correlation coefficient $\rho$ = −0.035 ± 0.03.

This analysis shows $FF = 0.5 \pm 0.02$ with a weak, but statistically significant, negative correlation $\rho = -0.035 \pm 0.03$, where $\pm$ denotes the 95% confidence intervals as obtained by bootstrapping (Fig 5C and 5D).

To infer parameters we first considered a purely deterministic approach of performing a least-square fit between the model-predicted dynamics of the mean QC and the data. Assuming constant values for the refilling probability $p_d$ and the release probability $p_r$, the model-predicted average QC over time (normalized by the first stimulus QC) is given as (11)

$$\frac{\langle \boldsymbol{b}_i \rangle}{\langle \boldsymbol{b}_1 \rangle} = \frac{\left( p_d + p_r \left(1 - p_r\right)^{i-1} \left(1 - p_d\right)^i \right)}{p_d + p_r(1 - p_d)}, \quad \forall i \in \{1, 2, \ldots\}. \tag{24}$$

Eq (24) shows a good fit to the synaptic depression observed in the electrophysiological results (Fig 6A) and results in the inferred values $p_r \approx 0.23$ & $p_d \approx 0.2$ suggesting that the synapses operate at low values for both these probabilities. It is noteworthy to contrast this estimate with the classical method of Elmqvist and Quastel (EQ) [54] that assumes no SV replenishment during the first 50 ms of a high-frequency challenge. Using this approximation, which as we will shortly see is violated in this case, yields an even lower $p_r \approx 0.12$ (using QCs from the first three stimuli).

How consistent are the inferred values of $p_r \approx 0.23$ & $p_d \approx 0.2$ obtained using least-square fitting with the steady-state QC fluctuation statistics? Interestingly, the obtained values for $p_r \approx 0.23$ & $p_d \approx 0.2$ are incompatible with the steady-state QC fluctuation statistics as reported in Fig 5. For example, using these values in the mathematical formulas from the previous section demonstrates a much higher model-predicted $FF$ of 0.87 and a stronger anticorrelation between QCs (Fig 6C). To be able to capture these steady-state statistics, one would need $p_r \approx 0.93$ & $p_d \approx 0.53$ as obtained by simultaneously solving Eqs (16) and (20) (model-predicted Fano factor and correlation, respectively) with the experimentally determined statistics of $FF = 0.5$ and $\rho = -0.035$. Bootstrapping the QC data from stimulus numbers 10 to 3000 to obtain $FF$ and $\rho$, and then solving Eqs (16) and (19) yields $p_r = 0.93 \pm 0.05$ & $p_d = 0.53 \pm 0.04$, where $\pm$ denotes the 95% confidence interval. These results show that MNTB-LSO synapses operate with a release probability and an SV refilling probability that are much higher than estimated by simply fitting the mean QC dynamics or by using the EQ method.

Fig 6B shows the predicted transient dynamics for these high constant probabilities $p_r = 0.93$ & $p_d = 0.53$. With these high values, the initial transient is much faster than the data (gray line in Fig 6B). Since the release probability is already quite high and close to one at steady-state (Fig 6C), we considered a model with a time-varying refilling probability to explain the observed short-term plasticity. More specifically, we considered different refilling probabilities $p_{d,1}, p_{d,2}, p_{d,3}, p_{d,4}, p_{d,5}$ for the first five stimuli, and then a fixed probability $p_d = p_{d,i}, \forall i \geq 6$ for the remaining stimuli. Recall that our proposed analytical framework provides an exact statistical QC distribution at each stimulus for such time-varying refilling probabilities. To infer these parameters we consider two alternative approaches. The first approach performs a least-square fit between the model-predicted mean QC and data, and this results in

$$p_{d,1} = 0.92, \; p_{d,2} = 0.73, \; p_{d,3} = 0.66, \; p_{d,4} = 0.53, \; p_{d,5} = 0.12 \; \& \; p_d = 0.51, \tag{25}$$

i.e., a refilling probability that starts high and then reaches its steady-state value within the first six stimuli (inset of Fig 6B plots the inferred refilling probabilities). In the alternative approach, we infer the parameter using a maximum likelihood approach that explicitly

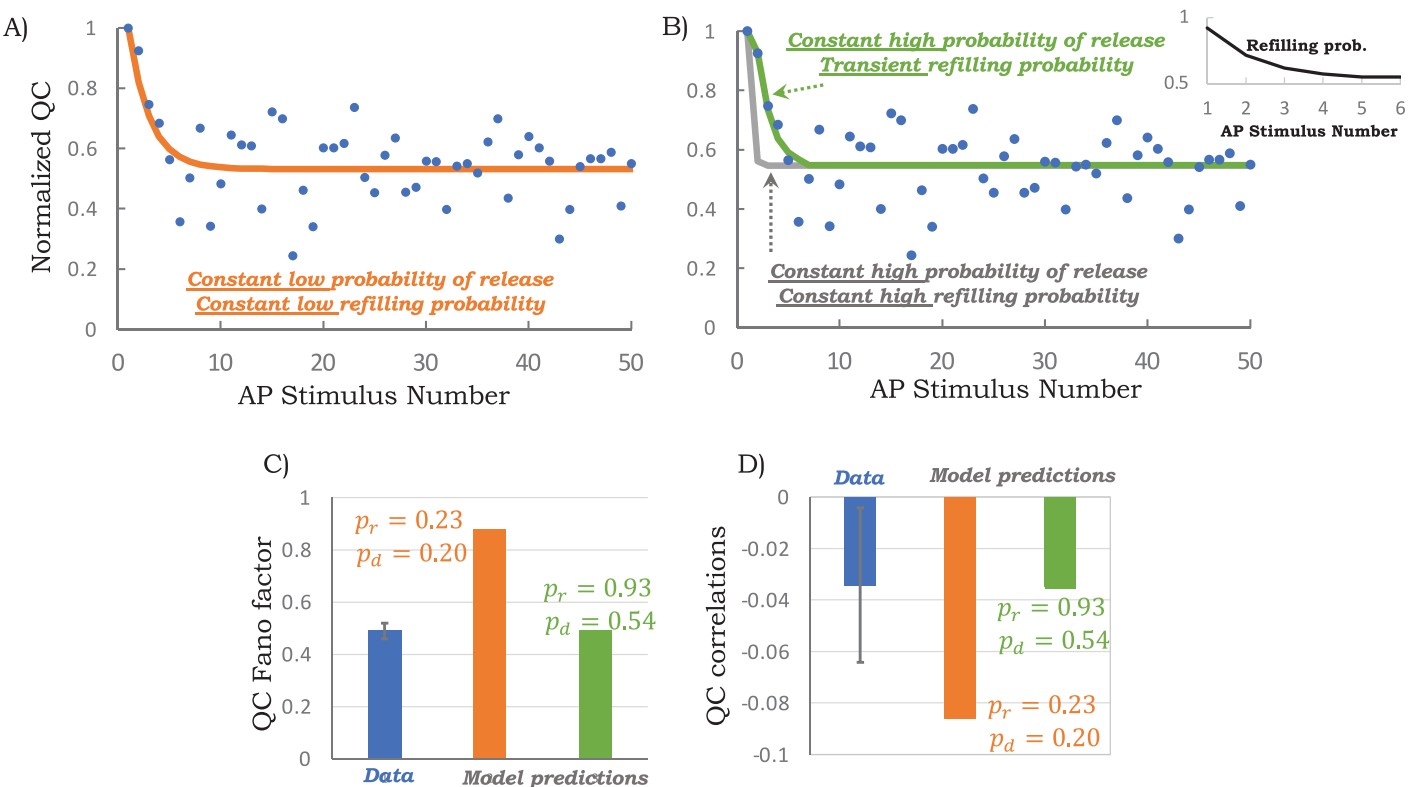

**Fig 6. The MNTB-LSO synapses are characterized by high refilling and release probabilities.** A) A quantitative fit of Eq (24) with $p_r = 0.23$ & $p_d = 0.2$ to the transient QC dynamics (orange line). B) Model-predicted QC dynamics as per Eq (24) with $p_r = 0.93$ & $p_d = 0.53$ (gray line), and as predicted by solving (2) for a constant release probability $p_r = 0.93$, zero undocking probability $p_{u,i} = 0$ and a time-varying refilling probability as shown in the inset (green line). C) Model-predicted steady-state QC Fano factors from (16) for constant low probabilities ($p_r = 0.23$ & $p_d = 0.2$ in orange) or high probabilities ($p_r = 0.93$ & $p_d = 0.53$ in green). Only the latter scenario is consistent with fluctuation statistics from the electrophysiological data (shown in blue). D) Model-predicted steady-state correlations between successive QCs from (20) for constant low (orange) and high probabilities (green), with only the latter fitting the experimentally obtained QC correlations (blue). Error bars on the data are the 95% confidence interval on the steady-state statistics as obtained from bootstrapping QCs from stimulus numbers 10 to 3000.

takes into account quantal size fluctuations and results in parameter values similar to (25) (Appendix F in S1 File). In summary, a transient refilling probability coupled with a high release probability explains both the synaptic depression characteristics and the steady-state QC fluctuation statistics (Fig 6B, green line).

## Discussion

We have investigated the stochastic dynamics of neurotransmission as governed by the depletion and replenishment of a single homogeneous readily-releasable pool of SVs in response to an AP train (Fig 1). The model is defined by a fixed number of docking sites $M$, where each site is characterized by three time-varying probabilities:

- The probability $p_{d,i}$ of an empty site becoming docked by a SV in the inter-stimulus interval. This probability is monotonically related to the time-varying kinetic rate of SV recruitment to empty sites (Appendix A).
- The probability $p_{u,i}$ of an occupied site becoming unoccupied during the inter-stimulus interval due to SV undocking or a spontaneous release event.
- The probability $p_{r,i}$ of AP-triggered SV fusion and neurotransmitter release.

We emphasize that the *sites operate independently and are identical in their parameters*. Our main contribution is the exact derivation of the QC transient statistics for this general class of models governing stochastic dynamics of neurotransmission. For a deterministic AP train with an inter-stimulus interval $1/f$, where $f$ is the frequency of stimulation, the number of readily-releasable SVs just before the $i^{th}$ AP is binomial with parameters $M$ and $p_i$ – each of the $M$ sites occupied with probability $p_i$, where $p_i$ is given as the solution to (2). Moreover, the transient QC distribution given by Eq (3) is also binomial with parameters $M$ and $p_i p_{r,i}$. As discussed further in Appendix C, this result can be generalized to scenarios where the inter-stimulus interval is random, in which case the QC distribution is non-binomial, and the QC Fano factor can exceed one.

Because of the transient distribution, for any stimulus within an AP train, the QC Fano factor for the $i^{th}$ AP can be directly related to the corresponding average QC (Eq (8) and Fig 2). Furthermore, alternative parameter regimes leading to the same average QC dynamics can be distinguished by their transient Fano factor profiles (Fig 3). The above assumption of identical sites can be relaxed by considering another set of sites $\tilde{M}$, with different parameters $\tilde{p}_{d,i}, \tilde{p}_{u,i}, \tilde{p}_{r,i}$. With two classes of docking sites (possibly resulting from differences in their proximity to calcium channels) – $M$ sites with parameters $\{p_{d,i}, p_{u,i}, p_{r,i}\}$, and $\tilde{M}$ sites with parameters $\{\tilde{p}_{d,i}, \tilde{p}_{u,i}, \tilde{p}_{r,i}\}$ – the transient QC is a sum of two binomially-distributed random variables $\boldsymbol{b}_i$ and $\tilde{\boldsymbol{b}}_i$ as given by (3) for their respective parameters. Thus, our analytical results can be generalized to consider heterogeneous SV pools operating in parallel, thus resulting in richer neurotransmission dynamics.

By further investigating the process during steady-state transmission, we derive exact formulas for the steady-state QC distribution, its associated Fano factor $FF$, the correlation coefficient $\rho$ between successive QCs, and QC autocorrelation function as a function of the steady-state refilling, undocking and release probabilities $p_d$, $p_u$ and $p_r$, respectively. In Appendix E and F these formulas are extended to consider quantal size fluctuations. Such steady-state statistics for synaptic transmissions have also been previously reported in the case of no undocking $p_u = 0$, and docking site refilling occurring as per a memoryless Poisson process [62,75,76]. As highlighted in Appendix A, the proposed modeling framework also relaxes this assumption by allowing the refilling/undocking rates in the inter-stimulus interval to vary arbitrarily with precise AP timing introducing some form of memory in these processes. As we point out later, relaxation of this memoryless assumption is key to understanding synaptic transmission with random inter-stimulus intervals.

Our analysis shows that high $(p_d, p_r \approx 1)$ as well as low $(p_d, p_r \ll 1)$ values of these probabilities lead to uncorrelated QCs (see the upper-left and lower-right regions of Fig 4B), these regions yield contrastingly different Fano factors, namely a $FF$ close to zero when both probabilities are high, yet a $FF$ close to one when both probabilities are low. Furthermore, intermediate value of these probabilities $(p_d, p_r \approx 0.5)$ leads to the most anticorrelated QCs (Fig 4). Using $FF$ and $\rho$ from electrophysiological data one can estimate $p_d$ and $p_r$ by simultaneously solving the nonlinear Eqs (16) and (20). For example, using a $FF = 0.5$ and $\rho = -0.035$ results in two sets of symmetric solutions: 1) $p_r = 0.93$ & $p_d = 0.53$ or 2) $p_r = 0.53$ & $p_d = 0.93$. The reason is that both (16) and (20) are themselves symmetric with respect to both $p_d$ and $p_r$. In this case, given the high refilling probability, the second solution ($p_r = 0.53$ & $p_d = 0.93$) predicts a normalized synaptic depression of 0.96 (steady-state mean QC normalized by the first stimulus). This is inconsistent with the synaptic depression observed in the data, which is closer to 0.55 (Fig 5). Thus, the first solution ($p_r = 0.93$ & $p_d = 0.53$) provides the physiologically relevant parameters that are consistent with the electrophysiological data in all three metrics: normalized synaptic depression, steady-state QC Fano factor, and QC correlation. It is important

to emphasize that knowing the mean QC and FF are by itself not sufficient to infer steady-state quantal parameters and knowledge of correlations is necessary to be able to uniquely infer $p_r$ and $p_d$. The approach of solving statistical lower-order moments (16) and (20) to infer $p_r$ and $p_d$ is analytically tractable and can be extended to consider quantal size fluctuations (Appendix E in S1 File). The parameters determined from here can be used in conjunction with computationally expensive likelihood-based approaches for parameter inference that explicitly take into account correlations between successive evoked PSCs [50].

The process of narrowing down the feasible parameter space is illustrated in Fig 7. Given the errors in quantifying QC from evoked PSCs, and other physiological sources of variation, such as differences in AP duration/amplitude [77], the *FF* obtained from electrophysiological data is an upper bound on the true QC Fano factor. To this end, one can mark a feasible parameter space consistent with $FF \leq 0.55$, where the conservative upper bound of 0.55 comes from taking a 10% range around the experimentally-observed $FF \approx 0.5$ in Fig 5C. The low *FF* value restricts the parameter space to the upper left corner (i.e., high values of $p_d$ and $p_r$) in the left-most plot in Fig 7. This can also be seen mathematically in Eq (18) where 1–*FF* provides a lower bound on both $p_d$ and $p_r$: the lower the *FF*, the higher the probabilities. While the results shown in this paper are based on a recording from a single postsynaptic LSO neuron, the analysis of 16 such recordings shows an $FF \leq 0.5$, in 75% of the cases (12 of 16), with some MNTB-LSO connections displaying a Fano factor as low as 0.2 (Fig C in Appendix G and data provided in S2 File).

By marking similar regions consistent with the observed QC correlations and normalized synaptic depression in Fig 7, and taking an intersection of these regions, the probabilities are constrained to the middle-left region of the parameter space (right plot in Fig 7), corresponding to a high release probability and a refilling probability in the range of $0.53 \pm 0.1$. To estimate the SV replenishment rate from the refilling probability we use Eq (27) in Appendix A with an inter-stimulus interval of 20 ms and $k_{u,i} = 0$ resulting in a recruitment rate of 37 SVs per empty site per sec. In this estimation procedure, we have assumed the undocking probability $p_u$ to be zero. For example, for $p_u = 0.2$, solving Eq (15) yields a refilling probability $p_d \approx 0.57$ which results in recruiting 54 SVs per empty site per sec. Finally, analysis of the transient depression in QCs seen in data (Fig 5B) indicates that the short-term plasticity is a result of a

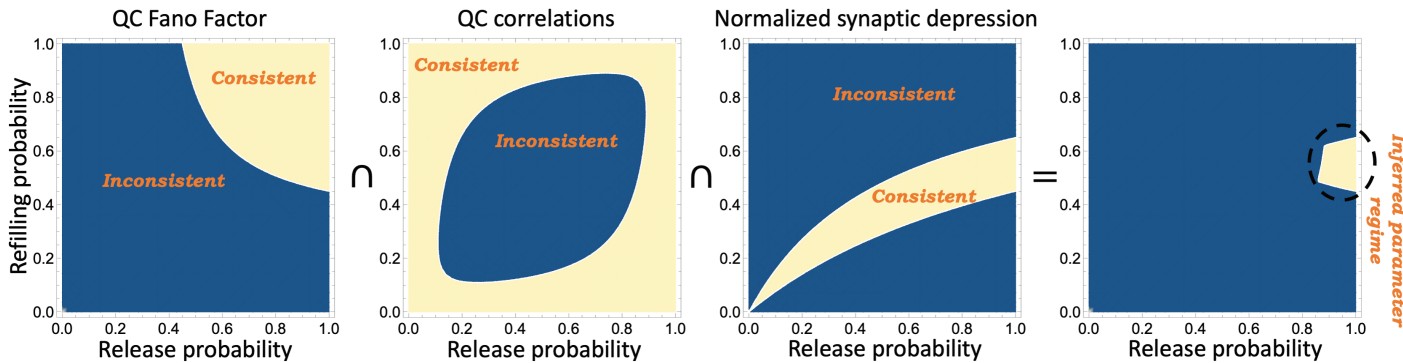

**Fig 7. Identification of release and refilling probabilities for MNTB-LSO synapses using QC fluctuation statistics**. The left-most-plot marks the region of parameter space (in terms of the refilling and release probabilities $p_d$ and $p_r$, respectively) consistent with a steady-state QC Fano factor as predicted by the formula (16) to be less than $FF \leq 0.55$. The *FF* upper bound is based on a 10% range around the experimentally-observed $FF \approx 0.5$ in Fig 5C. The other plots mark the parameter space consistent with QC correlations as given by (20) to be $\rho \geq -0.06$ and the normalized synaptic depression as given by (13) in the range $0.45 \leq \bar{p} \leq 0.65$ based on the electrophysiological data in Fig 5. The right-most-plot shows the intersection of all three consistent regions narrowing the parameter space to a region with a high release probability ($p_r \geq 0.85$) and $0.45 \leq p_d \leq 0.65$.

decreasing refilling probability (inset in Fig 6B). We plan to capture this phenomenon more mechanistically in future work by considering an SV pool that feeds into the readily releasable pool. It is interesting to note similarities between this model and the release-independent depression reported in [78] that is mediated by a decrease in the release probability to subsequent APs. However, it is important to note the key difference with the MNTB-LSO synapses, where the decrease occurs in the rate of refilling of empty docking sites with SVs.

In summary, the exact analytical solution for the transient QC distribution provides an elegant, novel framework to infer presynaptic model parameters from QC fluctuation statistics. We are currently working in close collaboration to test model predictions at the MNTB-LSO synapses for different frequencies and challenge durations. While in this manuscript we analyzed data from [69] based on a fixed inter-stimulus interval, it has been shown that variable inter-stimulus intervals (more specifically, a Poisson AP train) can provide more accurate parameter estimates [79]. In future work, we want to consider gamma-distributed inter-stimulus intervals where the noise in the intervals can be arbitrarily modulated. Preliminary data using this protocol on the MNTB-LSO synapses reveals SV refilling at docking sites occurring with rates that depend on the precise timing of the last AP, and this phenomenon can be incorporated into our general stochastic modeling framework with refilling occurring as per an inhomogeneous Poisson process (Appendix A).

We also plan to investigate other auditory synapses, such as glutamatergic calyx of Held-MNTB and Cochlear nucleus-LSO synapses [69], as well as synapses in the cerebellum [12,80] and the hippocampus [4]. On a theoretical level, we aim to extend these models to include loosely vs. tightly docked SVs as has been recently reported [81–83] and consider a repair period for docking sites before they become available for SV refilling [84,85]. Other avenues of future work involve exploring feedback control of neurotransmission, such as regulating presynaptic processes by secreted neurotransmitters through autoreceptors [86–90], and investigate stochastic dynamics of interconnected neurons starting with simple feedforward motifs [91,92].

## Supporting information

**S1 File. Appendix A–G containing detailed derivations of theoretical results.**
(PDF)
**S2 File. Microsoft Excel sheet with QC data from [69] used in Figs 5 and C in S1 File.**
(XLSX)

## Author contributions

**Conceptualization:** Abhyudai Singh.

**Data curation:** Oliver Gambrell, Abhyudai Singh.

**Formal analysis:** Zahra Vahdat, Oliver Gambrell, Abhyudai Singh.

**Funding acquisition:** Eckhard Friauf, Abhyudai Singh.

**Investigation:** Abhyudai Singh.

**Methodology:** Abhyudai Singh.

**Supervision:** Eckhard Friauf, Abhyudai Singh.

**Validation:** Jonas Fisch, Abhyudai Singh.

**Visualization:** Jonas Fisch, Abhyudai Singh.

**Writing – original draft:** Zahra Vahdat, Jonas Fisch, Eckhard Friauf, Abhyudai Singh.

**Writing – review & editing:** Jonas Fisch, Eckhard Friauf, Abhyudai Singh.

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
