## [Decision Letter · Decision Letter 0]

27 Nov 2024

PCOMPBIOL-D-24-01382Leveraging the transient statistics of quantal content to infer neuronal synaptic transmissionPLOS Computational Biology Dear Dr. Singh Thank you for submitting your manuscript to PLOS Computational Biology. After careful consideration, we feel that it has merit but does not fully meet PLOS Computational Biology's publication criteria as it currently stands. Therefore, we invite you to submit a revised version of the manuscript that addresses the points raised during the review process. Please submit your revised manuscript within 60 days Jan 27 2025 11:59PM. If you will need more time than this to complete your revisions, please reply to this message or contact the journal office at ploscompbiol@plos.org. Please include the following items when submitting your revised manuscript: * A rebuttal letter that responds to each point raised by the editor and reviewer(s). You should upload this letter as a separate file labeled 'Response to Reviewers'. This file does not need to include responses to formatting updates and technical items listed in the 'Journal Requirements' section below. * A marked-up copy of your manuscript that highlights changes made to the original version. You should upload this as a separate file labeled 'Revised Manuscript with Track Changes'. * An unmarked version of your revised paper without tracked changes. You should upload this as a separate file labeled 'Manuscript'. If you would like to make changes to your financial disclosure, competing interests statement, or data availability statement, please make these updates within the submission form at the time of resubmission. Guidelines for resubmitting your figure files are available below the reviewer comments at the end of this letter. We look forward to receiving your revised manuscript. Kind regards,Robert RosenbaumGuest EditorPLOS Computational Biology Andrea E. MartinSection EditorPLOS Computational Biology Feilim Mac GabhannEditor-in-ChiefPLOS Computational Biology Jason PapinEditor-in-ChiefPLOS Computational Biology **Additional Editor Comments :**  Both reviewers found the results to be interesting and clearly presented. However, in addition to minor concerns and questions, one reviewer had concerns about the novel contributions of the work. This reviewer asked that the authors clarify the novel contributions made by the manuscript, given previous work in these directions. **Journal Requirements:**

At this stage, the following Authors/Authors require contributions: Abhyudai Singh. Please ensure that the full contributions of each author are acknowledged in the "Add/Edit/Remove Authors" section of our submission form.

4) Your manuscript is missing the following sections: Results, and Methods.  Please ensure all required sections are present and in the correct order. Make sure section heading levels are clearly indicated in the manuscript text, and limit sub-sections to 3 heading levels. An outline of the required sections can be consulted in our submission guidelines here:

5) Please upload all main figures as separate Figure files in .tif or .eps format. For more information about how to convert and format your figure files please see our guidelines: 

6) We have noticed that you have uploaded Supporting Information files, but you have not included a list of legends. Please add a full list of legends for your Supporting Information files after the references list.

7) We notice that your supplementary Figures, and information are included in the manuscript file. Please remove them and upload them with the file type 'Supporting Information'. Please ensure that each Supporting Information file has a legend listed in the manuscript after the references list.

8) We note that your Data Availability Statement is currently as follows: "All relevant data are within the manuscript and its Supporting Information files.". Please confirm at this time whether or not your submission contains all raw data required to replicate the results of your study. Authors must share the “minimal data set” for their submission. PLOS defines the minimal data set to consist of the data required to replicate all study findings reported in the article, as well as related metadata and methods (https://journals.plos.org/plosone/s/data-availability#loc-minimal-data-set-definition).

9) Please amend your detailed Financial Disclosure statement. This is published with the article. It must therefore be completed in full sentences and contain the exact wording you wish to be published.

2) State what role the funders took in the study. If the funders had no role in your study, please state: "The funders had no role in study design, data collection and analysis, decision to publish, or preparation of the manuscript.".

**Reviewers' comments:**Reviewer's Responses to Questions

Reviewer #1: The authors derive a number of analytical results on the distribution of synaptic vesicles before and during vesicle release. They apply their results to published data on auditory synapses, which have several interesting properties due to their need for high-speed transmission. They find that the data is inconsistent with a constant rate of vesicle replenishment in this system.

The major weakness of this study is that it’s results are not placed in the context of existing results and this makes it very hard to disentangle the novel contributions that are made by the paper. For example:

* The fact that fluctuations are necessary to identify quantal parameters has been known since (at least) Bekkers (Curr Op Neurobiol, 1994) and has a well-known solution in mean-variance analysis (Silver, Momiyama, Cull-Candy, J Physiol 1998). Loebel, Silberberg et al (Front Comp Neuro, 2009) provides a nice example and the method is reviewed in Lanore and Silver (Neurometh, 2016). In this paper, none of this work is acknowledged and a very similar solution is introduced as if it were entirely novel (see for example italic emphasis at the end of section III).

* The introduction of time-varying parameters for vesicle release and recovery is interesting. Some mechanisms giving rise to this are discussed in Fuhrmann et al (J Physiol, 2004). However, the analytical results presented in the paper drop this assumption (either assuming time independence or steady-state behaviour), which means that the majority of equations in the Results appear to be special cases of results already derived in Goldman (Neural Comp, 2004), Rosenbaum, Rubin, and Doiron (PLoS CB, 2012) (both of which the manuscript cites), and in Bird and Richardson (PLoS CB, 2018) (which the manuscript doesn’t). It would be interesting to see exactly where this study has been able to identify new and significant results.

* The analysis of experimental data is also unsatisfying. The comparison to existing methods for the data analysis (above Fig 6) uses Elmqvist and Quastel (J Physiol, 1965), a paper almost 60 years old and does not reference any more modern methods. Further the authors neglect another potential source of variability in synaptic responses: variability in the response to a single vesicle. This has been acknowledged as a major problem since Kuno (J Physiol, 1964). Studies such as Turner and West (J Neurosci Meth, 1993) and Bhumbra and Beato (J Neurophysiol, 2013) have already applied mixture models to disentangle the effects of vesicle variability and quantal stochasticity for single pulses.

* The above methods have also been extended by Barri, Wang et al (eNeuro, 2016) and Bird, Wall, and Richardson (Front Comp Neuro, 2016) to account for synaptic plasticity. Both papers also explicitly leverage the serial correlations between quantal events to improve their estimates of the data, something that is again presented as a novelty here. It would be very interesting to see how the analysis conducted here compares to the more recently published literature. Bykowska, Gontier et al (Front Syn Neuro, 2019) provide a more recent review of methods that have been used to solve this problem.

* The general focus on fixed interspike intervals (periodic spike trains) is a weakness if the method is to be used for inference as Costa, Sjöström, and van Rossum (Front Comp Neuro, 2013) showed that variable (in particular uncorrelated exponential) interspike intervals are optimal for inferring the parameters of short-term plasticity.

* The finding that a time-varying replacement probability is necessary to fit the LSO data is interesting. Such a mechanism appears similar to the frequency-dependent recovery model introduced by Fuhrmann et al (J Physiol, 2004) and implemented as the FDR model in Bird, Wall, and Richardson (Front Comp Neuro, 2016). It would be particularly interesting if the authors could discuss this model and identify if time or frequency dependence is a more significant factor in vesicle recovery at these synapses.

Overall, although the authors have done good and careful work in this manuscript, the way it is written makes it very hard to identify a novel contribution to the field. If the manuscript could be extensively rewritten in the context of the literature mentioned above (and any other important papers I have missed) so that similarities with, differences from, and improvements over existing work are clearer, it would be much easier to assess the significance of the findings.

Reviewer #2: Understanding the dynamics of neurotransmitter filled synaptic vesicles (SVs) is important to understand interactions between neurons. These vesicles are docked at sites in the axon terminal and are released when an action potential (Aps) arrives at the axon terminal. The depletion of these SVs due to successive APs is counteracted by replenishment. These processes (release and replenishment/docking) are inherently probabilistic.

In this manuscript authors have built a probabilistic model of SV docking and release. They analytically derive the probability distribution of both occupation of sites and quantal content (QC), and show that it is binomial. They also derive formulas for fano factor (FF) and QC correlations (ρ). They show that different parameter regimes that produce the same average behaviour can be distinguished based FF and ρ. They apply their results to electrophysiological data obtained from 50 Hz stimulation of MNTB-LSO synapse. They show that the method of Elmqvist and Quastel explains the mean synaptic depression but not the fluctuations in QC. They combine information from average synaptic depression, FF, and ρ to infer model parameters that better explain the fluctuation statistics of the QC.

Overall this is a very interesting paper which not only provides a biologically interpretable model of vesicle docking and release but authors also provide an easy scheme to infer model parameters from experimental data. In principle I support the publication of the paper. There are only a few points where I think authors need to provide more clarification or possibly extend their model:

Authors have only studied synaptic depression. How the results would be for synaptic facilitation. Authors should include at least one example of parameter setting that would lead to synaptic facilitation

The equations are symmetric for p_d and p_r. Is there a reason for considering only the former to vary with time and not the latter?

The authors do not comment on the frequency dependence of synaptic plasticity in their model. In appendix A they relate the probabilities to rate kinetics and time period of stimulus, but do not mention how these rate kinetics themselves depend on frequency of the input. I think this is an important issue that needs to be elaborated.

**Have the authors made all data and (if applicable) computational code underlying the findings in their manuscript fully available?**

Reviewer #1: Yes

Reviewer #2: **No: **The code for the calculations and model is not made available

PLOS authors have the option to publish the peer review history of their article (what does this mean?). If published, this will include your full peer review and any attached files.

Reviewer #1: No

Reviewer #2: No

**Figure resubmission:** While revising your submission, please upload your figure files to the Preflight Analysis and Conversion Engine (PACE) digital diagnostic tool, https://pacev2.apexcovantage.com/. PACE helps ensure that figures meet PLOS requirements. To use PACE, you must first register as a user. Registration is free. Then, login and navigate to the UPLOAD tab, where you will find detailed instructions on how to use the tool. If you encounter any issues or have any questions when using PACE, please email PLOS at figures@plos.org. Please note that Supporting Information files do not need this step. If there are other versions of figure files still present in your submission file inventory at resubmission, please replace them with the PACE-processed versions. 
---

## [Decision Letter · Decision Letter 1]

18 Apr 2025

Dear Dr. Singh,

We are pleased to inform you that your manuscript 'Interfering synaptic transmission from the stochastic dynamics of the quantal content: An analytical approach' has been provisionally accepted for publication in PLOS Computational Biology.

Best regards,

Robert Rosenbaum

Guest Editor

PLOS Computational Biology

Andrea E. Martin

Section Editor

PLOS Computational Biology

Both reviewers were satisfied with the changes to the manuscript. One reviewer had minor suggestions for edits. Please consider these suggestions when submitting your final files.

Reviewer's Responses to Questions

**Comments to the Authors:**

Reviewer #1: I am satisfied with the changes made by the authors in the revision process and find the revised manuscript suitable for publication as it places the careful work done into the context of the existing literature. I do have some further comments on the text:

1) There is probably a typo in the title (‘Interfering’…)

2) I don’t think that the last sentence of the Abstract is particularly relevant as it does not focus on the novel contributions of this paper.

3) Line 12: ‘much work … deterministic’. I think the phrasing here is unfair; these studies are aware that release is stochastic but only study the average behaviour.

Reviewer #2: I am happy with the revision. Authors have addressed my concerns and i do not have further comments.

**Have the authors made all data and (if applicable) computational code underlying the findings in their manuscript fully available?**

Reviewer #1: Yes

Reviewer #2: Yes

PLOS authors have the option to publish the peer review history of their article (what does this mean?). If published, this will include your full peer review and any attached files.

Reviewer #1: No

Reviewer #2: No

---

## [Editor Report · Acceptance letter]

PCOMPBIOL-D-24-01382R1

Inferring synaptic transmission from the stochastic dynamics of the quantal content: An analytical approach

Dear Dr Singh,

I am pleased to inform you that your manuscript has been formally accepted for publication in PLOS Computational Biology. Your manuscript is now with our production department and you will be notified of the publication date in due course.

With kind regards,

Anita Estes
